# Guidance for reporting intervention development studies in health research (GUIDED): an evidence-based consensus study

Edward Duncan [1] , Alicia O'Cathain [2] Nikki Rousseau,[1,3] Liz Croot,[2] Katie Sworn,[4] Katrina M Turner [5] Lucy Yardley,[6,7] Pat Hoddinott [1]

For numbered affiliations see end of article.

**Correspondence to**
Dr Edward Duncan;
edward.duncan@stir.ac.uk

## ABSTRACT

**Objective** To improve the quality and consistency of intervention development reporting in health research.

**Design** This was a consensus exercise consisting of two simultaneous and identical three-round e-Delphi studies (one with experts in intervention development and one with wider stakeholders including funders, journal editors and public involvement members), followed by a consensus workshop. Delphi items were systematically derived from two preceding systematic reviews and a qualitative interview study.

**Participants** Intervention developers (n=26) and wider stakeholders (n=18) from the UK, North America and Europe participated in separate e-Delphi studies. Intervention developers (n=13) and wider stakeholders (n=13) participated in a 1-day consensus workshop.

**Results** e-Delphi participants achieved consensus on 15 reporting items. Following feedback from the consensus meeting, the final inclusion and wording of 14 items with description and explanations for each item were agreed. Items focus on context, purpose, target population, approaches, evidence, theory, guiding principles, stakeholder contribution, changes in content or format during the development process, required changes for subgroups, continuing uncertainties, and open access publication. They form the GUIDED (GUIDance for the rEporting of intervention Development) checklist, which contains a description and explanation of each item, alongside examples of good reporting.

**Conclusions** Consensus-based reporting guidance for intervention development in health research is now available for publishers and researchers to use. GUIDED has the potential to lead to greater transparency, and enhance quality and improve learning about intervention development research and practice.

## BACKGROUND

The UK Medical Research Council's (MRC) Framework for the Development and Evaluation of Complex Interventions[1] lists intervention development as the first of a series of interconnected phases. While the MRC Complex Intervention Guidance has stimulated considerable methodological progress in understanding and reporting the latter phases (ie, feasibility and piloting, evaluation, and implementation), the intervention development phase has remained relatively underdeveloped and without a comprehensive reporting guideline.[2] Research funders, researchers, commissioners, practitioners, the public and patients are increasingly interested in understanding and improving the intervention development process.

There are a variety of ways to develop interventions. A review of approaches include partnership (eg, coproduction; codesign), target population-centred, evidence-based and theory-based, implementation-based, efficiency-based, step-based or phased-based, intervention-specific, or a combination of methods.[3] Successful intervention development is characterised as being rigorous, scientific and resulting in effective interventions that can be implemented in real-world settings.[4] However, a key intervention development challenge is the lack of evidence-based quality criteria on which to assess which, if any, approach is superior to another and in which context.

The reasons why intervention development processes are currently under-reported are

### Strengths and limitations of this study

► The items were developed through a structured and transparent consensus-based process.
► Parity of opinion was given to intervention developers and wider stakeholders throughout the development of the reporting guidance.
► Despite aiming to secure an international sample, this proved difficult.
► We acknowledge that participants in the study were predominantly based in the Global North, and that the perspectives of intervention developers and wider stakeholders from the Global South are absent.

unclear. This may be due to research funding priorities or pressure to publish efficacy or effectiveness studies, diminishing the priority of publishing intervention development studies. When intervention development studies are published, they are sometimes included as part of a feasibility or pilot study. Consequently, detail about how the intervention was actually developed can be sparse. A more systematic, comprehensive and transparent approach to intervention development reporting is likely to enhance understanding about the intervention development process. It would help readers to understand the benefits and challenges of different intervention development approaches. It would help researchers select an intervention development approach that is relevant to their context. It would also facilitate future retrospective assessment of how different intervention development approaches can lead to either effective or ineffective interventions that do or do not translate into practice change. Potentially such assessment could provide insights into research waste. While some reporting guidance already exists that relates to intervention development, these are limited in scope. The Template for Intervention Description and Replication (TIDiER)[5] guidelines are extensions to Consolidated Standards of Reporting Trials (CONSORT) for improving the reporting of the completed intervention that results from the intervention development process. The Criteria for Reporting the Development and Evaluation of Complex Interventions in healthcare: revised guideline (CReDECI 2e) does provide reporting guidance for intervention development; however, this is limited to four items, as CREDECI 2 also provides guidance on reporting feasibility and piloting and evaluation. To date, there has been no guidance focusing in detail on reporting the whole process of intervention development.

This paper presents the GUIDance for the rEporting of intervention Development (GUIDED). GUIDED forms part of a larger MRC-funded study to produce guidance on intervention development: the IdentifyiNg and assessing different approaches to DEveloping compleX interventions (INDEX) study.[6] It is the first international mixed-methods consensus study to focus on reporting guidance solely for intervention development. In this paper we report the methods used to develop and gain consensus on the items included in the GUIDED checklist. We present each reporting item with further description and explanation. GUIDED will be of interest to research funders, researchers, journal editors, commissioners, practitioners, the public and patients, who we refer to collectively as 'readers'.

## METHODS
### Design
We published our intent to develop intervention reporting guidance (5 July 2017) in the EQUATOR (Enhancing the QUAlity and Transparency Of Health Research) Network Library (http://www.equator-network.org/library/ reporting-guidelines-under-development/reporting-guidelines-under-development-for-other-study-designs/#80). The design of this intervention development consensus study involved conducting two simultaneous and identical e-Delphi studies, followed by a consensus workshop. Participants included (1) intervention developers and (2) wider stakeholders who were involved in the wider intervention development activities, including directors of research funding panels, editors of journals that had published intervention development studies, public and patient involvement members of intervention development studies, and people working in health service implementation.[6] By separating intervention developers and wider stakeholders within the e-Delphi process, we ensured that the perceptions of both groups were equally reported and their views given equal weight. A subset of the consensus exercise related specifically to the identification of intervention development reporting guidance, reported in this paper. We followed established methods for developing reporting guidance[7] (see figure 1) and report the e-Delphi guidance in line with current best practice.[8] The parallel e-Delphi studies were delivered over three separate rounds. Each round lasted for 4 weeks. Non-responders were emailed a reminder after 2 weeks. Completion of one round was required to enter the next e-Delphi round. There was space for participants to comment beside each item and explain their responses, or (in round 1) suggest alternative item wording. Comments were reviewed by the research team, but were not sufficient in number or depth to require analysis using a formal method. No additional items were suggested by participants and no changes were made to the wording of items between rounds. Items were not removed from subsequent rounds, even if they had previously passed the predetermined threshold.

### e-Delphi item generation
e-Delphi items were generated by triangulating three different data sources: a systematic methods overview of 87 articles, books and websites that identified 23 approaches to intervention development within 8 categories and with 18 actions undertaken across these approaches[3]; a systematic review of 87 international primary research articles reporting intervention development processes which describes 10 actions[9]; and an analysis of 21 indepth qualitative interviews with an international sample of intervention developers (n=15) and key stakeholders (n=6).[4 10] The research team (AOC, LY, PH, ED, LC, NR, KS) met regularly to identify the potential reporting guidance items. Members of the research team worked in pairs, and one team of three, to extract potential guidance items from the three data sources. Each pair then presented potential e-Delphi items to the whole team. Each potential item was discussed, refined and agreed. We grouped items into themes, with one theme entitled 'Reporting Guidance', which had 18 items. The full set of e-Delphi items and their ratings have been reported elsewhere.[6]

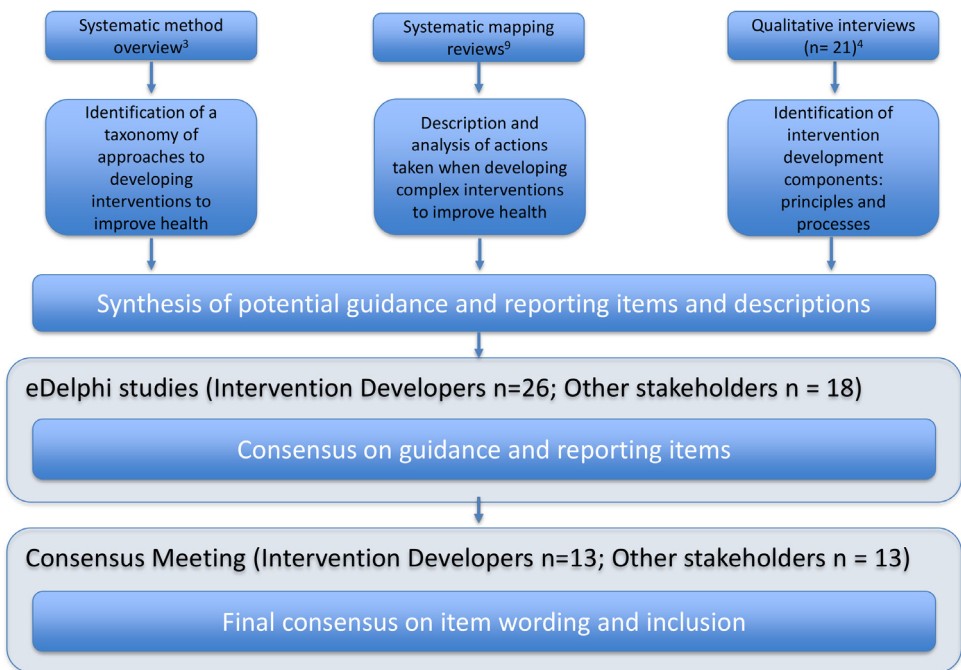

**Figure 1** Methods used to develop reporting guidance for intervention development studies in health research.

### e-Delphi participants

Invitations were sent to 92 individuals who had undertaken intervention development and/or published a formal approach to intervention development and 80 wider stakeholders. Intervention developers, identified through parallel studies conducted by the research team,[3 9] were invited to participate if they had published at least one intervention development study or written methodological books or journal articles about intervention development. Wider stakeholders were identified through a web search of journal editorial boards, funding bodies and other relevant sources. Wider stakeholders were invited if their role brought them into direct contact with the intervention development process, for example as editors, funding panel members (patient and public involvement members and subject experts) or commissioners. In addition, we convened an international expert panel with members from the UK, USA and Europe early in the project to guide the research.[6] Approximately one-third of invited participants were from countries other than the UK. Members of this expert panel participated in the e-Delphi studies and consensus workshop alongside other participants. Individuals who responded to say they would participate in the e-Delphi were emailed a study information containing a URL to an established e-Delphi platform[11] and a unique password to access the study.

### Definition of consensus

Following an online consent process, participants were asked to rate the importance they would give to each potential item, when developing complex interventions to improve health, on a scale of 1–5: not at all important (1); slightly unimportant (2); somewhat important (3); fairly important (4); very important (5). An additional option of no relevant expertise was provided for each item. In our grant application, we decided that an item would be included within the reporting guidance if at least 70% of participants agreed that an item was fairly important (4) or very important (5) in either e-Delphi group by the end of round 3. We discussed this with our expert panel prior to undertaking the e-Delphi. Including items that reached the predefined threshold in either group meant that equal priority was given to participants that belonged to either the intervention development or wider stakeholder group. A similar approach to methodological guideline reporting development has been used elsewhere.[12]

### Consensus meeting

The results of the e-Delphi studies were discussed by participants (in person or by video link) and eight team members at a 1-day consensus meeting on 13 March 2018 in London, UK. The meeting began with presentations from the team: an overview of the overall INDEX study, followed by a summary overview of both systematic reviews[3 9] and qualitative study.[4] The results of both e-Delphi studies were then presented, and detailed discussions were held on items that had not reached consensus, but which reflected divergence. Consensus meeting participants did not rerate items but suggested how the wording of some items may have caused confusion. We took this into consideration in our interpretation of the e-Delphi. However, this did not affect any of the reporting items.

### Following the consensus meeting

After the meeting, two of the reporting items were merged in the final GUIDED checklist, as they were considered by

the team to be appropriately covered in a single item: item 10 'Report how the intervention changed in content and format from the start of the intervention development process' and a recommendation to 'report the reasons for discarding intervention components that were considered'. No further changes were made to reporting guidance items. We identified examples of previous studies that illustrate use of the reporting items. These examples are almost completely drawn from reviews of previous intervention development literature.[4 9] In one instance, where an example could not be found, an example was created and is identified as a hypothetical example.

### Patient and public involvement

Patients and public individuals (PPI) were invited (1) to an expert panel meeting held at the start of the study, (2) to participate in the wider stakeholder e-Delphi study and (3) to participate in the consensus meeting. None attended the first meeting, but PPI members did participate in the wider stakeholder e-Delphi study and did attend the consensus conference aimed at interpreting the Delphi results. All study participants will be sent copies of journal publications resulting from the study.

## RESULTS
### Description of participants

The response rates for each round were as follows: round 1 intervention developers: n=34, wider stakeholders: n=22; round 2 intervention developers: n=27, wider stakeholders: n=18; and round 3 intervention developers: n=26, wider stakeholders: n=18. Intervention development participants who completed round 3 were based in the UK (n=16), mainland Europe (n=5), Ireland (n=4) and USA (n=1). They included people from public health (n=10), applied health research/health services research (n=8), psychology (n=7), nursing (n=6) and allied health professional (n=1) backgrounds. Wider stakeholder participants who completed round 3 were based in the UK (n=16), mainland Europe (n=1) and USA (n=1). They included chairs or members of funding panels (n=5), editors or editorial board members of journals (n=4), commissioners of services (n=3), public and patient involvement (n=3), and other (n=3) individuals.

The 26 participants of the consensus meeting were based in the UK (n=19), USA (n=3), mainland Europe (n=3) and Ireland (n=1). They were invited due to their varied roles in the intervention development process: intervention developers (n=13), methodologists (n=4), chairs of funding panel (n=3), journal editors (n=3), public and patient representatives (n=1), commissioner (n=1), and other (n=1).

### Description of consensus from e-Delphi study

Fifteen of a possible 18 intervention development reporting items reached our a priori threshold for inclusion.[6] Table 1 presents all the reporting items included in the e-Delphi, the percentage of responses that scored 4

or 5 for each item and the mode score. The responses to round 3 of the full Delphi study are available in the full guidance.[6]

### GUIDED intervention development reporting items: description and explanation

Below, we have ordered the items so that those which are more likely to be considered earlier in the development process are listed first. However, there is no fixed order in which the reporting items must be considered. The EQUATOR reporting guidance[7] encourages describing and explaining the rationale for each reporting item to help researchers and others to write or appraise reports. We have therefore followed this format, in keeping with other related reporting guidelines.[5 13 14]

A blank checklist to support the use of GUIDED by authors and reviewers is provided in online supplementary file 1.

### Item 1

*Description:* Report the context for which the intervention was developed.

*Explanation:* Understanding the context in which an intervention was developed informs readers about the suitability and transferability of the intervention to the context in which they are considering evaluating, adapting or using the intervention. Context here can include place and organisational and wider sociopolitical factors that may influence the development and/or delivery of the intervention.[15]

#### Example

1a. The purpose of this article is to describe the development of a theory-based, culturally relevant intervention focusing on primary (sexual risk reduction) and secondary (Pap smear) prevention of cervical cancer *among Latina immigrants* using intervention mapping (IM).[16]

1b. In an effort to bridge this evidence-practice gap, we have developed a behaviour change intervention that aims to increase provision of upper limb repetitive task-oriented training in *stroke rehabilitation.*[17]

### Item 2

*Description:* Report the purpose of the intervention development process.

*Explanation:* Clearly describing the purpose of the intervention specifies what it sets out to achieve. The purpose may be informed by research priorities, for example those identified in systematic reviews, evidence gaps set out in practice guidance such as the National Institute for Health and Care Excellence, or specific prioritisation exercises such as those undertaken with patients and practitioners through the James Lind Alliance.

#### Example

2a. The aim of this phase of the project was *to develop a CBTi that could be delivered by* HCAs (non-specialist,

**Table 1** e-Delphi study results for intervention development reporting items across each e-Delphi round.

| Items | Intervention developers | | | Wider stakeholder | | |
|---|---|---|---|---|---|---|
| | Mode score (% agreement by round) | | | | | |
| | Round 1 | Round 2 | Round 3 | Round 1 | Round 2 | Round 3 |
| Report the purpose of the intervention. | 5 (97) | 5 (96) | 5 (100) | 5 (94) | 5 (100) | 5 (100) |
| Report the target population. | 5 (97) | 5 (96) | 5 (100) | 5 (100) | 5 (100) | 5 (100) |
| Report any use of components from an existing intervention. | 5 (84) | 5 (89) | 5 (100) | 4 (89) | 4 (93) | 4 (100) |
| Report how evidence from different sources informed the intervention development. | 5 (93) | 5 (96) | 5 (100) | 5 (83) | 5 (86) | 5 (100) |
| Report how stakeholders contributed to the intervention development process. | 5 (97) | 5 (96) | 5 (100) | 4 (89) | 4 (93) | 4 (94) |
| Report important uncertainties at the end of the intervention development process. | 5 (87) | 5 (93) | 5 (100) | 5 (83) | 5 (86) | 5 (78) |
| Report the context for which the intervention was developed. | 5 (90) | 5 (93) | 5 (96) | 5 (94) | 5 (93) | 5 (100) |
| Report any changes to interventions required or likely to be required for subgroups. | 5 (90) | 5 (89) | 5 (96) | 5 (83) | 4 (93) | 4 (83) |
| Report how any published intervention development approach contributed to the development process. | 5 (83) | 5 (78) | 5 (92) | 4 (67) | 4 (64) | 4 (71) |
| Report how existing published theory informed the intervention development process. | 5 (87) | 5 (89) | 5 (92) | 4 (89) | 5 (93) | 4 (94) |
| Report any guiding principles, people or factors which were prioritised when making decisions. | 5 (81) | 5 (85) | 5 (92) | 4 (72) | 4 (93) | 4 (83) |
| Report how the intervention changed in content and format from the start of the intervention development process.* | 5 (74) | 4 (74) | 5 (88) | 4 (77) | 4 (93) | 4 (94) |
| Report the reasons for discarding intervention components that were considered.* | 5 (74) | 5 (81) | 5 (88) | 4 (78) | 4 (93) | 4 (88) |
| Follow TIDieR guidance when describing the developed intervention. | 5 (76) | 5 (69) | 5 (80) | 4 (100) | 5 (100) | 5 (88) |
| Report the intervention development in an open access format (eg, open access journal, report chapter, website). | 5 (68) | 4 (67) | 4 (77) | 5 (77) | 5 (86) | 5 (89) |
| Report the background and contribution of those making decisions about the intervention content, format and delivery. | 5 (50) | 3 (40) | 3 (42) | 4 (61) | 4 (67) | 4 (67) |
| Report the time taken to develop the intervention. | 4 (52) | 3 (41) | 3 (27) | 3 (33) | 3 (21) | 3 (17) |
| Report who, when, why and where the original idea for developing the intervention came from. | 3 (45) | 3 (30) | 3 (27) | 5 (50) | 4 (64) | 4 (67) |

Items not reaching the threshold for inclusion in round three are shaded.[38]
*These items were merged into one item (see item 10) following the consensus meeting.
TIDieR, Template for Intervention Description and Replication.

relatively low-paid staff), *after training in basic CBT skills.*[18]

## Item 3

*Description:* Report the target population for the intervention development process.

*Explanation:* The target population is the population that will potentially benefit from the intervention—this may include patients, clinicians and/or members of the public. If the target population is clearly described, then readers will be able to understand the relevance of the intervention to their own research or practice. Health inequalities, gender and ethnicity are features of the target population that may be relevant to intervention development processes.

### Example

3a. The purposes of this study were to use patient-centered, mixed-methods intervention development techniques to develop and refine a computer-based intervention for *drug-using women reporting IPV [intimate partner violence] in the ED [emergency department]…* The intervention was developed in a 5-step process: (1) Initial intervention development based on selected theoretical frameworks; (2) *In-depth interviews with the target population*; (3) Intervention adaptation, with iterative feedback from further interviews; (4) Beta testing and review by an advisory committee of domestic violence advocates; (5) Acceptability and feasibility testing in a small open trial.[19]

## Item 4

*Description:* Report how any published intervention development approach contributed to the development process.

*Explanation:* Many formal intervention development approaches exist and are used to guide the intervention development process (eg, six steps in quality intervention development (6SQuID)[20] or the person-based approach to intervention development[21]). Where a formal intervention development approach is used, it is helpful to describe the process that was followed, including any deviations. More general approaches to intervention development also exist and have been categorised as follows[3]: target population-centred intervention development; evidence-based and theory-based intervention development; partnership intervention development; implementation-based intervention development; efficacy-based intervention development; step-based or phased-based intervention development; and intervention-specific intervention development.[3] These approaches do not always have specific guidance that describe their use. Nevertheless, it is helpful to give a rich description of how any published approach was operationalised.

### Example

4a. *We applied each of the six 6SQuID steps in relation to five areas of adolescent health-related behaviour* (substance

use, sleep, sexual behaviour, physical activity, and eating behaviour), in order to determine how our review findings could further enhance the development of effective, relevant, and sustainable interventions. This involved incorporating the specific neurological changes that happen during adolescence, such as the alterations that occur in the limbic system and pre-frontal cortex, and their influence on areas such as decision-making and reward processing [7]. The steps outlined in Table 1 were therefore considered in relation to adolescent physiology and health-related behaviour, and the results indicate their application to intervention development.[22]

4b. *Our approach was based on the key development activities outlined in the MRC framework for the development and evaluation of complex interventions.*[89] There *were four non-sequential activities,* which: A. Identified the existing evidence on interventions to modify explanatory factors for HRQoL (symptoms and ability to function in a normal role and social environment). B. Identified and developed theory to underpin the intervention by convening an expert group to discuss the evidence in the context of expert knowledge and wider theory. C. Modelled processes and outcomes by reconvening the expert group to identify theory-based behaviour change techniques (BCTs), conceptualise the theoretical mechanisms of change in the intervention and identify process and outcome evaluation measures. D. Assessed intervention feasibility and acceptability through interviews with patients who might receive the intervention and health professionals who would deliver it. It was then piloted with a small group of patients.[23]

## Item 5

*Description:* Report how evidence from different sources informed the intervention development process.

*Explanation:* Intervention development is often based on published evidence and/or primary data that have been collected to inform the intervention development process. It is useful to describe and reference all forms of evidence and data that have informed the development of the intervention because evidence bases can change rapidly, and to explain the manner in which the evidence and/or data were used. Understanding what evidence was and was not available at the time of intervention development can help readers to assess transferability to their current situation.

### Example

5a. *The three methods used will be realist literature review, secondary analysis of the English Longitudinal Study of Ageing (ELSA), and qualitative focus groups and interviews.… A realist approach will be taken to synthesise and integrate data.*[34] *This theory will be explored with stakeholders to develop an intervention which will be tested and refined in a feasibility trial.*[24]

5b. *The [mixed lit review and qualitative interview]*

*findings were discussed with a stakeholder panel to agree the components of an intervention.* Stakeholders included stroke researchers, health professionals and service user representatives from the King's College London Stroke Research Patients and Family Group. Findings were initially presented to stakeholders who were asked to consider in light of the evidence, the types of intervention that might be feasible. The larger group then reconvened for group discussion to co-design an intervention. As a peer support model had been proposed, we scoped this literature to identify likely mechanisms of change that could operate as 'active ingredients' of peer support to improve resilient practices after stroke. We then modelled process and outcomes to develop a theoretically informed intervention to promote resilience after stroke.[24]

## Item 6

*Description:* Report how/if existing published theory informed the intervention development process.

*Explanation:* Reporting whether and how theory informed the intervention development process aids the reader's understanding of the theoretical rationale that underpins the intervention. Although not mentioned in the e-Delphi or consensus meeting, it became increasingly apparent through the development of our guidance that this item could relate to either existing published theory or programme theory.

### Example

6a. *The content and structure of the video were designed in line with best practice in health psychology and behavior change theory, and recent qualitative research into psychosocial aspects of alcohol use* (Brown & Gregg, 2012; de Visser et al., 2013; Jayne et al., 2010; Livingstone et al., 2011; MacNeela & Bredin, 2011; Szmigin et al., 2011). *The approach involved several different techniques identified in Abraham and Michie's (2008) taxonomy of behavior change intervention techniques* -providing information about others' approval; prompting identification of barriers; providing general encouragement; modeling behavior; and providing opportunities for social comparison; providing information about consequences; teaching use of prompts/cues; and planning social support. Employed.[25]

6b. *The framework driving the design of this multimedia intervention is the chronic care self-management model (CCM).* The CCM transforms a reactive health care system into one that improves patient outcomes through planning, proven strategies, management, and patient activation [38, 39]. The model results in healthier patients, more satisfied providers, and lower health care utilizations and can be applied to cancer surgical populations [40, 41]. The CCM recognizes six essential elements of high-quality care. These include (1) systems that promote safe, high-quality care; (2) effective, efficient clinical care that includes patient and FCG self-management support;

(3) care that is evidence-based and family-centered; (4) efficient and effective care through organized data; (5) care that empowers and prepares patients and FCGs to manage healthcare; and (6) inclusion of community resources [41]. The six essential elements and responsive intervention content are described in Table 1.[26]

## Item 7

*Description:* Report any use of components from an existing intervention in the current intervention development process.

*Explanation:* Some interventions are developed with components that have been adopted from existing interventions. Clearly identifying components that have been adopted or adapted and acknowledging their original source helps the reader to understand and distinguish between the novel and adopted components of the new intervention.

### Example

7a. I-DECIDE also *builds on the weave project* (Hegarty, O'Doherty, Gunn, Pierce, & Taft, 2008), a randomized, controlled trial that evaluated the effectiveness of a face-to-face counselling intervention in the primary care setting for women experiencing fear of a partner (Hegarty et al., 2013b). The doctors in the intervention group were provided with training in the delivery of a counselling intervention involving woman centered care, active listening, motivational interviewing techniques, and nondirective problem solving to validate and respond to the woman's experiences and feelings (Hegarty et al., 2013b). *The I-DECIDE intervention aims to translate aspects of the weave counselling intervention, namely, tailored responses and messaging and the use of motivational interviewing and nondirective problem solving tools into an online format.* It can thus be seen as a form of therapeutic intervention, which differentiates I-DECIDE from the other similar websites being developed in the United States, Canada, and New Zealand, which are primarily decision aids.[27]

7b. *Rather than creating an entirely new intervention, it will be developed by adapting a promising seizure management course that already exists to address the needs of PWE visiting the ED.* The course titled 'Epilepsy awareness and seizure management' has been offered on a small scale within the third sector to people from a variety of backgrounds, including patients, teachers and care home staff by the UK charity, Epilepsy Society. The society has offered the course since 1998 and given us permission to adapt it. It has not been formally evaluated, but aims to increase participants' confidence in seizure management. Changes to the existing course will be required since it was developed for delivery to a narrower, fee-paying group. It was not created for delivery within the health service, nor for PWE who visit EDs who can be particularly challenged by

epilepsy and may have lower education.[9 17 27 36 37.][28]

## Item 8

*Description:* Report any guiding principles, people or factors that were prioritised when making decisions during the intervention development process.

*Explanation:* Reporting any guiding principles that governed the development of the intervention will help the reader to understand the authors' reasoning behind the decisions that were made. Guiding principles specify the core objectives and features of the desired intervention.[29] These could include prioritising patient preferences over clinician preferences, providing an engaging experience for patients, minimising the cost of delivering the intervention, or maximising the potential for the intervention to be scaled up.

### Example

8a. …the value of parental participation in pediatric weight loss trials has previously been demonstrated (21) but not reinforced in studies to date. How the involvement of a caregiver influences weight outcomes in minority youth remains an understudied area (9, 22). *It was hypothesized that greater caregiver involvement in treatment would be associated with greater weight loss.*[30]

8b. Offering a DV [domestic violence] intervention in an online format may assist in overcoming some of the barriers encountered in health care settings. Online interventions are being increasingly used as a way of self-managing health conditions, with promising results (Murray, Burns, See Tai, Lai, & Nazareth, 2005). Lintvedt et al. (2013) point out *that an Internet-based intervention is constantly available and accessible from any location. This flexibility allows women to access the intervention at unexpected times when an abusive partner is not present, as opposed to the health care setting where they must schedule an appointment. Delivering an intervention online also allows women to self identify and self-manage without disclosure to a third party.* This may be particularly beneficial for women who are unable or unwilling to disclose the abuse to a health care professional and would not attend a specialized support service because they do not perceive themselves as a 'DV victim' (Zink, Elder, Jacobson, & Klostermann, 2004).[27]

## Item 9

*Description:* Report how stakeholders contributed to the intervention development process.

*Explanation:* Potential stakeholders can include patient and community representatives, local and national policy makers, healthcare providers, and those paying for or commissioning healthcare. Each of these groups may influence the intervention development process in different ways. Specifying how differing groups of stakeholders contributed to the intervention development process helps the reader to understand how stakeholders

were involved and the degree of influence they had on the overall process. Further details on how to integrate stakeholder contributions within intervention reporting are available.[31]

### Example

9a. A key element of the intervention development process was patient and public involvement (PPI) [22]. The REACH-HF programme has an active PPI group consisting of six people from Cornwall with a range of experiences of heart failure and three caregivers of people with heart failure… *The REACH-HF PPI group helped to design the topic guide for the focus group interviews, they completed and commented on the needs assessment survey and commented on summaries of information from the focus groups.* The group met every 2 months throughout the 12 month needs assessment stage with additional e-mail and postal correspondence between meetings.[32]

9b. *We used a qualitative approach in two stages* (Fig. 2), informed by prestudy pilot work with clinicians and interviews with three service users with experience of sharing news of a cancer diagnosis to guide development of the study methodology. *User interviews informed the study in three ways*: (i) the timing of recruitment, (ii) purposive sampling of participants and (iii) interview guide development.[33]

## Item 10

*Description:* Report how the intervention changed in content and format from the start of the intervention development process.

*Explanation:* Due to the iterative nature of intervention development, the intervention that is defined at the end of the development process can often be quite different from the one that was initially planned. Describing these changes and their rationale enhances understanding and enables other intervention developers to learn from this experience. For example, it may be that some intervention components were considered but ultimately discarded due to complexity or expense of delivery.

### Example

Intervention changes may be substantial (eg, changing the mode of delivery of an intervention from face-to-face to online delivery) or relatively more subtle as illustrated in the following:

10a. Our usability evaluation not only assessed the users' ability to perform tasks successfully on the DEF website but also examined the users' broader interaction with the website comprising their thoughts, feelings, and perceptions [103]. The method used to execute this step was a first experience for the research team and was based on regular practice in the field of usability evaluation [103,104]. Detailed activities of this step are described below…… *Following this step, numerous improvements (ie, more than 40) were made, ranging from minor (eg, changing a word that was hard to*

*understand on a page to an easier one) to major (eg, redesigning a whole page).* Here are some of the major improvements made to the website: (1) reduction of the number of pages that must be viewed to complete the enrollment process (ie, 5 pages fewer, from 20 to 15 pages), (2) replacement of the neutral background of videos to backgrounds inspiring PA behavior, (3) redesign and reorganization of the information on the self-monitoring tool page, (4) reduction of the length of the information messages and summary messages by about 50% for almost every tailored motivational session, (5) significant reduction in the number of choices for MI-based multiple choice questions in tailored motivational sessions 3 and 5, (6) redesign of the emails sent to recruit participants, and (7) reduction in the number of times participants have to scroll to almost zero.[34]

10b. *The text messages developed by different stakeholders in this study were substantially revised by caregivers participating in focus group discussions conducted to test understanding of the messages.* The discussions revealed ambiguities, assumptions and unfamiliar terms used in the messages. Terms like 'AL' and 'reassessment' were replaced with words that were more general and easy to understand. Similarly, a message that required a caregiver to take the child to the hospital immediately if unwell was revised to take into consideration circumstances like lack of transport if a child fell sick at night. *Messages that could have multiple interpretations were identified and revised.* Participants of FGDs preferred personalized SMS messages addressing the caregiver by their specific name. This informed the decision to programme the automated distribution system to send each message with a salutation addressing the caregiver by their name and language of choice. Using their specific names made the caregivers feel that the message was addressing them personally, even though the messages were not tailored to their individual needs.[35]

### Item 11

*Description:* Report any changes to interventions required or likely to be required for subgroups.

*Explanation:* Specifying any changes that the intervention development team perceive are required for the intervention to be delivered or tailored to specific subgroups enables readers to understand the applicability of the intervention to their target population or context. These changes could include changes to personnel delivering the intervention, to the content of the intervention or to the mode of delivery of the intervention.

### Example

11a. There is the potential to expand this intervention to a wider group of health care professionals involved within cancer care such as oncologists, radiologists, dieticians, speech and language therapists, physiotherapists, occupational therapists and those working in primary care such as GPs, health care assistants and community nurses. If the behavioural diagnosis can be confirmed for these groups then the intervention can be deemed to be feasible for these audiences. There is the potential to adapt this intervention to encourage physical activity in other long-term condition patients by health care professionals. Again, *if the behavioural diagnosis can be confirmed, the structure of the intervention, the selected intervention functions, policy categories, BCTs and modes of delivery could be deemed feasible, with changes made to the intervention content. Further, if the behavioural diagnosis can be confirmed for delivery of physical activity advice by health care professionals to those identified as inactive within the general population, then the intervention structure could again be deemed feasible, with changes made to the intervention content.* In this instance however, the suggested 'teachable moment'[16,20] might not be present and therefore changes might be less likely to occur in the physical activity behaviours of those in receipt of advice.[36]

### Item 12

*Description:* Report important uncertainties at the end of the intervention development process.

*Explanation:* Intervention development is frequently an iterative process. The conclusion of the initial phase of intervention development does not necessarily mean that all uncertainties have been addressed. It is helpful to list remaining uncertainties such as the intervention intensity, mode of delivery, materials, procedures or type of location that the intervention is most suitable for. This can guide other researchers to potential future areas of research and practitioners about uncertainties relevant to their healthcare context.

### Example

No good-quality illustration of this item could be found in the literature. The following hypothetical example is offered to illustrate how this item could be reported.

12a. At the conclusion of the intervention development process, we were able to describe the rationale and theoretical basis for the intervention in detail. We were also able to list the materials that are required to deliver the intervention and the training materials that are required to train staff in its use. The intervention should be delivered face to face and provided individually. *While it may be possible for the number of intervention sessions to be personalised depending on the severity of participant's condition, we did not investigate participant or carer acceptability of a variable intervention intensity delivery approach.*

### Item 13

*Description:* Follow TIDieR guidance when describing the developed intervention.

*Explanation:* Interventions have been poorly reported for a number of years. In response to this, internationally recognised guidance has been published to support the high-quality reporting of healthcare interventions[5]

and public health interventions.[13] This guidance should therefore be followed when describing a developed intervention.

*Example*

13a. *Using the TIDieR framework [21], we created a broad outline of the intervention* that included the content delivered, to whom and by whom, why, by what mode of delivery and how often. Data from all three data sources were used.[37]

13b. *The 'Template for Intervention Description and Replication' (TIDieR) checklist [16] has been used to structure the description of this intervention.* The 12 checklist items are displayed in Table A (available online), alongside how each has been interpreted for this intervention description.[38]

### Item 14

*Description:* Report the intervention development process in an open access format.

*Explanation:* Unless reports of intervention development are available, people considering using an intervention cannot understand the process that was undertaken and make a judgement about its appropriateness to their context. It also limits cumulative learning about intervention development methodology and observed consequences at later evaluation, translation and implementation stages. Reporting intervention development in an open access (gold or green) publishing format increases the accessibility and visibility of intervention development research and makes it more likely to be read and used. Potential platforms for open access publication of intervention development include open access journal publications, freely accessible funder reports or a study web page that details the intervention development process.

*Example*

14a. Corder *et al.*[39]

14b. Free *et al.*[40]

### DISCUSSION

Intervention development is a vital component of the MRC developing and evaluating complex interventions guidance. This study presents a mixed-method international consensus study to produce detailed reporting guidance for the intervention development phase of the MRC Complex Intervention Guidance. The GUIDED checklist provides a list of 14 intervention development items, each with an accompanying explanation for why it is important to include this information in publications and outputs that describe the intervention development process. The GUIDED checklist was developed in collaboration with a range of stakeholders, each of whom contributed a range of expertise and perspectives on the intervention development process. Despite efforts to include participation from a global audience, the majority of participants come

from within the UK. Among developers, we had a good response from European countries but a poorer response from the rest of the world. Among wider stakeholders, the response was poor from all outside the UK. It seems likely that the study was more relevant to developers and that developers were also more likely to know of the study team, which may have influenced participation. To maximise response, any similar research in the future may benefit from a preliminary email endorsement from an influential person based in the same geographical region as the intended participant.

The GUIDED reporting checklist and its associated item descriptions have been systematically developed to support readers to understand key aspects of specific intervention development studies. Adhering to the GUIDED item checklist across the variety of formats in which intervention development publications already occur should improve the quality, transparency and consistency of intervention development reporting.

### What gap does GUIDED fill?

Good-quality effectiveness studies with detailed guidance on intervention description are necessary.[5 13] GUIDED is offered as complementary reporting guidance to detail the intervention development process. Presenting intervention development studies in line with GUIDED recommendations reported in this paper will enable commissioners and practitioners to understand the context and methods that were used to develop the intervention to help them make judgements about the quality and relevance of the intervention. This information will be useful in guiding their decisions about whether to evaluate or implement an intervention within their specific context. Finally, high-quality and transparent reporting of intervention development in line with GUIDED recommendations will enable methodological lessons to be learnt and incorporated into future intervention development studies. We therefore recommend that authors follow GUIDED when reporting intervention development studies, and journal editors and research funders endorse the use of GUIDED within any publications that report intervention development studies. The GUIDED checklist will be placed on the EQUATOR Network website, and we request that journals provide links to the EQUATOR site and signpost potential authors to this guidance where appropriate.

### How does GUIDED fit with other reporting guidance?

GUIDED provides a more comprehensive description than previous guidance[41] of what should be reported when publishing intervention development studies. GUIDED complements and can easily be integrated or signposted to within other reporting guidance. Papers that are written to describe interventions should follow existing guidance[5 13] and signpost readers to where they can read about the intervention development process, reported in line with GUIDED recommendations, so they can judge the appropriateness of the intervention

development process. Where randomised controlled trials are being reported using CONSORT guidance,[14] authors could signpost (eg, in reporting CONSORT Statement 5: Intervention) to where a GUIDED description of intervention development has been reported. Where patients and the public contributed to intervention development, the Guidance for Reporting Involvement of Patients and the Public (GRIPP 2) guidelines can be used.[31]

## CONCLUSION

The GUIDED checklist and reporting guidance has been developed by following internationally recognised methods for developing reporting guidance,[7] with items based on extensive primary[4 10] and secondary[3 9] research to enable greater transparency and quality of reporting development of complex interventions. The GUIDED checklist and guidance provides a clear and structured basis for the reporting of intervention development studies in a range of formats. It has the potential to facilitate learning about how early intervention development decisions impact across the life history of an intervention: through feasibility and efficacy testing, cost-effectiveness evaluations, and translation into healthcare practice change.

**Author affiliations**
¹Nursing, Midwifery and Allied Health Professional Research Unit, University of Stirling, Stirling, UK
²School of Health and Related Research, University of Sheffield, Sheffield, UK
³Leeds Institute of Clinical Trials, School of Medicine, University of Leeds, Leeds, UK
⁴Health Economics and Decision Science, School of Health and Related Research, University of Sheffield, Sheffield, UK
⁵Bristol Medical School, University of Bristol, Bristol, UK
⁶Psychology, University of Southampton, Southampton, UK
⁷School of Health Sciences, University of Bristol, Bristol, UK

**Acknowledgements** We would like to thank all the e-Delphi study and consensus conference participants and members of our expert panel.

**Contributors** AOC and PH led the development of the overall intervention development guidance. ED led the consensus exercise working with NR. ED wrote the first draft of the article and integrated contributions from the author group into subsequent drafts. All authors contributed to the design and content of the reporting guidance and approved subsequent drafts of the paper (AOC, PH, LY, LC, NR, KMT, ED, KS).

**Funding** The study was funded by the MRC-NIHR Methodology Research Panel (MR/N015339/1). Funders had no influence on the guidance presented here. The authors were fully independent of the funders.

**Competing interests** None declared.

**Patient and public involvement** Patients and/or the public were involved in the design, or conduct, or reporting, or dissemination plans of this research. Refer to the Methods section for further details.

**Patient consent for publication** Not required.

**Provenance and peer review** Not commissioned; externally peer reviewed.

**Data availability statement** De-identified participant response data are available from edward.duncan@stir.ac.uk upon reasonable request.

**ORCID iDs**
Edward Duncan http://orcid.org/0000-0002-3400-905X
Alicia O'Cathain http://orcid.org/0000-0003-4033-506X
Katrina M Turner http://orcid.org/0000-0002-6375-2918
Pat Hoddinott http://orcid.org/0000-0002-4372-9681

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
