## [Reviewer comments · BMJ Open]

ARTICLE DETAILS

TITLE (PROVISIONAL)	Guidance for reporting intervention development studies in health research (GUIDED): An evidence-based consensus study.
AUTHORS	Duncan, Edward; O'Cathain, Alicia; Rousseau, Nikki; Sworn, Katie; Turner, Katrina; Yardley, Lucy; Hoddinott, Pat

VERSION 1 – REVIEW

REVIEWER	Lawrence Mbuagbaw McMaster University
REVIEW RETURNED	19-Aug-2019

GENERAL COMMENTS	The investigators describe the processes employed to develop the GUIDED checklist - a reporting guideline for intervention development in health research. The paper is well written and clear. As the investigators state, the absence of participation from the “global south” is an important limitation. This is unfortunate given the possibility of reaching stakeholders in all parts of the world using the e-Delphi approach. What informed the threshold of consensus (70% rate item as fairly important)? The exemplified outlines in supplementary file 2 would be ideal in the main body of the manuscript. It helps understanding to have the example immediately after the explanation. Comment on alternative consensus approaches and why the Delphi (eDelphi) was chosen.
---

REVIEWER	Sean Grant Indiana University Richard M. Fairbanks School of Public Health, Indianapolis, IN, United States
REVIEW RETURNED	09-Dec-2019

GENERAL COMMENTS	I read "Reporting GUIDance for intervEntion Development in health research (GUIDED)" for consideration at BMJ Open. This manuscript addresses an important gap in reporting guidance for intervention research, and the work informing this guidance followed best practices in reporting guideline development. I offer some thoughts for the authors to consider in a revision:  - e-Delphi participants: Please provide details on how the initial list of 92 intervention developers and 80 wider stakeholders was identified. - e-Delphi participants: Please also provide operational definitions for each group of "wider stakeholders" akin to the detail provided for intervention developers. - Definition of consensus: The authors state that participants rated the importance of items, though it's not clear the importance "for
--

	what"? More details in the narrative and a copy of each round's questionnaire as a supplement would be helpful to provide.  - Definition of consensus: The authors state that the definition of consensus was determined prior to commencement. To allow external verification, the authors should provide a link to or copy of their protocol. - Definition of consensus: Please provide the rationale for the "70% ratings of 4 or 5 in either group" operationalization of consensus used. The authors also do not provide any details on the plan used to analyze the qualitative rationale comments provided by participants alongside their ratings. - Consensus meeting: Please provide details on how consensus was developed in the meeting (e.g, was a rating form used here as well?). - New section after "Consensus meeting": Please provide details on how the reporting guidance was developed/revised/finalized after the consensus meeting. - Patient and Public Involvement: How were PPI members identified, and how many were approached? - Description of participants: The response rates provided narratively would be better and easier to understand as a flow diagram. - Description of participants: This sentence fits better in the next section, as it is not about participants but the items "All fourteen intervention development reporting items reached our a priori threshold for inclusion (Table 1)(7)." - Description of consensus from survey: In addition to the final round results, the authors should provide detail on the number of items in each round and any important changes in ratings across rounds. As the paper currently reads, it seems as though all items had high agreement, leaving the reader to wonder why three rounds was necessary. - Description and explanation: Please provide a direct link to the "full copy of the guidance with checklist is available" rather than the generic link for the EQUATOR Network (www.equator-network.org). - Item 7: The authors should consider adding a recommendation to use established taxonomies for intervention components (e.g., the Behavior Change Technique Taxonomy). - Item 13: The authors should consider adding repositories (e.g., the Open Science Framework). - Discussion: The difficulty in finding interested stakeholders outside the UK is surprising. Could the authors critically reflect on the approaches that they used to identify, recruit, and retain stakeholders to offer some potential explanations as to WHY they struggled to recruit individuals outside of the UK? - Supplementary File 12: If the authors cannot find a good example in the literature, they should consider creating one (and flagging us such) to provide something for readers to model.
--	---

VERSION 1 – AUTHOR RESPONSE

REVIEWER 1

The paper is well written and clear.

Thank you. We have worked to improve the the clarity further through responding to both reviewers' comments.

As the investigators state, the absence of participation from the “global south” is an important limitation. This is unfortunate given the possibility of reaching stakeholders in all parts of the world using the e-Delphi approach.

We agree. Around a third of the potential participants we approached were from outside the UK. Among developers, we had a good response from European countries but a poorer response from the rest of the world. Among wider stakeholders, the response was poor from all the sample outside the UK. It seems likely that the survey was more salient to intervention developers and that developers were also more likely to know of the INDEX team. We have included further detail to highlight our attempt to recruit an international sample on page 4.

What informed the threshold of consensus (70% rate item as fairly important)?

There is no standard threshold for consensus in guideline development, nor in Delphi studies more generally. Previous studies using Delphi methods have ranged between a 50% and a 100% threshold^[1]. The research team discussed the acceptable threshold level in team meetings and consulted with our international expert panel. It was agreed, prior to commencement of the eDelphi study, that at least 70% of participants agreeing an item was fairly important (4) or very important (5) in either e-Delphi group by the end of Round 3, was sufficient to deem there was adequate agreement about the importance of an item.

The exemplified outlines in supplementary file 2 would be ideal in the main body of the manuscript. It helps understanding to have the example immediately after the explanation.

We agree and have inserted the examples immediately after the explanation as requested.

Comment on alternative consensus approaches and why the Delphi (eDelphi) was chosen.

We used a Delphi method as it is recommended best practice in the internationally recognised guidance for developers of Health Research reporting Guidelines^[2]. We cite this guidance in our Methods section on page 4. Given the established convention and acceptance of using Delphi methods for this purpose, we have not added further discussion on alternative approaches.

REVIEWER 2

e-Delphi participants: Please provide details on how the initial list of 92 intervention developers and 80 wider stakeholders was identified.

Further detail is provided on page 4 of the revised paper.

e-Delphi participants: Please also provide operational definitions for each group of "wider stakeholders" akin to the detail provided for intervention developers.

Operational definitions of wider stakeholders are provided on pages 4 and 6 of the paper. We hope this is sufficient.

Definition of consensus: The authors state that participants rated the importance of items, though it's not clear the importance "for what"? More details in the narrative and a copy of each round's questionnaire as a supplement would be helpful to provide.

To clarify the issue of importance, we have added, “when conducting high quality intervention development and reporting” to the text on page 5.

The questions asked in each round were identical. A list of all items and participants’ responses in Round 3 have been published by our team. We have now referred to this and a supporting reference given on page 6. We do not feel it is helpful to reproduce this in the reporting guidance paper as it

contains items that do not pertain to reporting guidance and may cause confusion. All reporting items are listed in the paper in Table 1.

Definition of consensus: The authors state that the definition of consensus was determined prior to commencement. To allow external verification, the authors should provide a link to or copy of their protocol.

There is no published protocol. The threshold was proposed in our original (unpublished) grant application and confirmed through consultation with our international expert panel prior to the commencement of the eDelphi study.

Definition of consensus: Please provide the rationale for the "70% ratings of 4 or 5 in either group" operationalization of consensus used. The authors also do not provide any details on the plan used to analyze the qualitative rationale comments provided by participants alongside their ratings.

As described above in response to Reviewer 1, there is no standard threshold for consensus in guideline development, nor in Delphi studies more generally. Previous studies using Delphi methods have ranged between a 50% and a 100% threshold^[2]. The research team discussed the acceptable threshold level in team meetings and consulted with international expert panel. It was agreed, prior to commencement of the eDelphi study, that that at least 70% of participants agreeing an item was fairly important (4) or very important (5) in either e-Delphi group by the end of Round 3, was sufficient to deem there was adequate agreement about the importance of an item.

Including items that reached the pre-defined threshold in either group meant that equal priority was given to participants that belonged to either the intervention development or wider stakeholder group. A similar approach has been used in another methodological guideline development study.^[4] We have added further detail to this effect on page 5.

Qualitative comments were reviewed by the research team, but were not sufficient in number or depth to require analysis using a formal method. Additional detail to this effect have been added on page 4

Consensus meeting: Please provide details on how consensus was developed in the meeting (e.g, was a rating form used here as well?).

We have added further detail to the consensus meeting section as requested. Participants at the consensus meeting suggested alternative wording of some items (not reporting items). This was taken into account in our interpretation of the eDelphi findings. Agreement was measured by discussion and consensus within the meeting. We did not use rating forms to assess this.

New section after "Consensus meeting": Please provide details on how the reporting guidance was developed/revised/finalized after the consensus meeting.

A new section has been added on page 5, as requested.

Patient and Public Involvement: How were PPI members identified, and how many were approached?

As described above, we have included further detail on wider stakeholders, on pages 4-5 of the revised paper. Wider stakeholders, including PPI, were identified through a web search of Journal editorial boards, funding bodies, and other relevant sources. They were invited if their role brought them into direct contact with the intervention development process, for example as editors, funding panel members, or commissioners. We do not have a list of the number of wider stakeholders from each group we invited.

Description of participants: The response rates provided narratively would be better and easier to understand as a flow diagram.

Our paper is developed and reported in keeping with MOHER guidelines^[5] and internationally respected and highly cited reporting guideline papers^[6]. None of these reporting guidelines include the flow diagram requested. We do not believe its inclusion within the current submission is warranted.

Description of participants: This sentence fits better in the next section, as it is not about participants but the items "All fourteen intervention development reporting items reached our a priori threshold for inclusion (Table 1)(7)."

We agree, and have moved this sentence as suggested.

Description of consensus from survey: In addition to the final round results, the authors should provide detail on the number of items in each round and any important changes in ratings across rounds. As the paper currently reads, it seems as though all items had high agreement, leaving the reader to wonder why three rounds was necessary.

Table 1 has now been updated to provide detail on the results for each item across each of the rounds, for both groups. In revising Table 1 we became aware of an editorial error in our original submission. There were in fact 18 Reporting Items in the study, of which 15 reached consensus. This has been clarified in the text and additions have been made to the paper and supplementary file as required. We apologise for this error. We are confident that the figures now reported are correct.

Description and explanation: Please provide a direct link to the "full copy of the guidance with checklist is available" rather than the generic link for the EQUATOR Network (www.equator-network.org).

We have deleted this sentence, as we will need the paper to be published prior to submission to the EQUATOR Network. The full guidance is however included in the paper and supplementary files.

Item 7: The authors should consider adding a recommendation to use established taxonomies for intervention components (e.g., the Behavior Change Technique Taxonomy).

We have considered this, but feel that we need to focus only on the items used in the e-Delphi, and only on items relevant to all approaches to intervention development.

Item 13: The authors should consider adding repositories (e.g., the Open Science Framework).

We have considered this, but feel that we need to focus only on the items used in the eDelphi. Participants were not asked about the Open Science Framework and other repositories.

Discussion: The difficulty in finding interested stakeholders outside the UK is surprising. Could the authors critically reflect on the approaches that they used to identify, recruit, and retain stakeholders to offer some potential explanations as to WHY they struggled to recruit individuals outside of the UK?

As discussed above, around a third of the potential participants we approached were from outside the UK. Among developers, we had a good response from European countries but a poorer response from the rest of the world. Among wider stakeholders, the response was poor from all the sample outside the UK. It seems likely that the survey was more salient to intervention developers. It may be that people who recognised the names of the research team were more likely to respond, and that research team members are better known in the Global North, than the Global South. We have included further detail to highlight our attempt to recruit an international sample on page 4 and have added further discussion on page 11

Supplementary File 2: If the authors cannot find a good example in the literature, they should consider creating one (and flagging us such) to provide something for readers to model.

Thank you for this suggestion. We have added a hypothetical example and clearly indicated it as such (See page 14).

VERSION 2 – REVIEW

REVIEWER	Sean Grant Richard M. Fairbanks School of Public Health Indiana University, Indianapolis United States
REVIEW RETURNED	10-Jan-2020
GENERAL COMMENTS	Thank you for your time and efforts.